# Highly efficient field-free switching of perpendicular yttrium iron garnet with collinear spin current

Man Yang [1,4], Liang Sun [1,4], Yulun Zeng[1], Jun Cheng [1], Kang He[1], Xi Yang[1], Ziqiang Wang[1], Longqian Yu[1], Heng Niu [1], Tongzhou Ji[1], Gong Chen [1], Bingfeng Miao [1] ✉, Xiangrong Wang [2,3] ✉ & Haifeng Ding [1] ✉

Yttrium iron garnet, a material possessing ultralow magnetic damping and extraordinarily long magnon diffusion length, is the most widely studied magnetic insulator in spintronics and magnonics. Field-free electrical control of perpendicular yttrium iron garnet magnetization with considerable efficiency is highly desired for excellent device performance. Here, we demonstrate such an accomplishment with a collinear spin current, whose spin polarization and propagation direction are both perpendicular to the interface. Remarkably, the field-free magnetization switching is achieved not only with a heavy-metal-free material, Permalloy, but also with a higher efficiency as compared with a typical heavy metal, Pt. Combined with the direct and inverse effect measurements, we ascribe the collinear spin current to the anomalous spin Hall effect in Permalloy. Our findings provide a new insight into spin current generation in Permalloy and open an avenue in spintronic devices.

Magnetic insulators can transport spin angular momentum without accompanying any charge carrier, showing great potential in low-power spintronic and magnonic applications. Among various magnetic insulators, yttrium iron garnet (YIG) is the "miracle material" as it has a millimeter-long magnon diffusion length and the lowest damping of any material reported so far[1,2] Traditionally, it has been widely used in various microwave devices, including phase shifters[3], isolators[4,5], and circulators[6], etc. Recent years have also witnessed a resurgence of interest in YIG for its strong potential in spintronic and magnonic applications. The successful demonstrations of fast domain wall motion[7], magnetization switching/oscillation with low current densities[8,9], and information transfer with high efficiency[10,11] make YIG a promising candidate for magnetic logic and memory devices.

To increase the integration capabilities of the devices and reduce power consumption, the field-free electrical control of perpendicular YIG magnetization with considerable efficiency is indispensable. YIG is a magnetic insulator, in which charge current cannot flow. Thus, the spin transfer torque effect, which was successfully demonstrated and used in spin-valves and magnetic tunnel junctions[12], cannot be applied[8]. The current-induced spin–orbit torque (SOT) naturally emerges as the best option since the pure spin current can be injected into the magnetic insulator[13]. The conventional method of generating pure spin current, such as the spin Hall effect (SHE), is limited by the orthogonality constraints among the spin polarization, propagation direction of the spin current, and the charge current direction[14]. Thus, the spin current from the conventional method does not support the deterministic switching of the perpendicular magnetization (Fig. 1a). And the aid of an additional magnetic field is typically required. This, however, is not ideal for device miniaturization and power consumption minimization. The field-free switching of the perpendicular magnetization is hotly pursued and has been realized by incorporating spin current sources with specially treated materials[15–43]. These treatments, such as special single crystalline materials, high temperature deposition, shape engineering, composition gradient control etc., are not

[1]National Laboratory of Solid State Microstructures, Department of Physics, Nanjing University, and Collaborative Innovation Center of Advanced Microstructures, Nanjing 210093, P.R. China. [2]Physics Department, The Hongkong University of Science and Technology, Clear Water Bay, Kowloon, Hongkong. [3]HKUST Shenzhen Research Institute, Shenzhen 518057, P.R. China. [4]These authors contributed equally: Man Yang, Liang Sun. ✉e-mail: bfmiao@nju.edu.cn; phxwan@ust.hk; hfding@nju.edu.cn

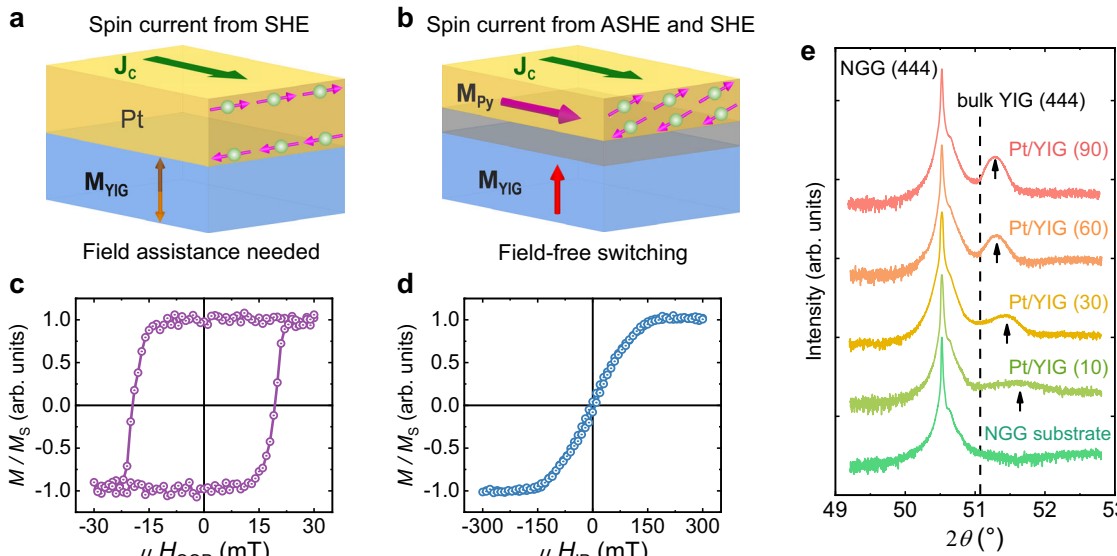

**Fig. 1 | Schematic illustration of SOT-switching and the magnetic/structural properties of the YIG films. a** Illustration of a conventional SOT-switching caused by spin Hall effect in PMA-YIG/Pt. In this configuration, the YIG film has the same stability for magnetization up and down with the same injecting current. Thus, an additional in-plane field is required to achieve the deterministic switching. **b** Illustration of a field-free SOT-switching by the combination of both anomalous spin Hall effect and spin Hall effect in PMA-YIG/Ag/Py. **c**, **d** are the out-of-plane (OOP) and in-plane (IP) hysteresis loops of 10-nm PMA-YIG measured with VSM, respectively. **e** X-ray diffraction data of a bare NGG substrate and YIG films with different thicknesses grown on NGG. The dashed line represents the bulk YIG (444) peak, and the unit of thickness is nm. The thickness of the Pt buffer layer is 0.5 nm.

industrial-friendly. Alternatively, a recent theory predicted that the symmetry constraint can also be removed for the spin current generation in an ordinary ferromagnetic metal. In an effect termed the anomalous spin Hall effect (ASHE)[44], a collinear spin current can be generated. With both the spin polarization and propagation direction perpendicular to the interface, this spin current may provide an opportunity to achieve the deterministic switching of perpendicular YIG (Fig. 1b). The experimental realization has yet to be reported.

In this work, we demonstrated the current-induced field-free switching of the perpendicular YIG magnetization utilizing the collinear spin current generated in an in-plane magnetized Permalloy (Py, $Ni_{80}Fe_{20}$) layer. The 6-nm Py layer is magnetically decoupled from the 10-nm thick YIG via a 4-nm Ag layer. The reasons for material selection are discussed in Supplementary Note 1. We utilize the thermoelectric measurements to detect the anomalous inverse spin Hall effect (AISHE), namely, the spin-to-charge conversion in Py induced by the collinear spin current generated in YIG. It exhibits a feature that can be modulated by both the YIG and Py magnetization, consistent with the scenario of AISHE. The deterministic field-free switching of the perpendicular YIG is further demonstrated via magneto-optical Kerr effect (MOKE) measurements. It shows a Py magnetization-controlled feature, in good agreement with the theoretical prediction. Remarkably, field-free switching is achieved not only with a heavy-metal-free material but also with a higher efficiency since the current density threshold is only about half of that used in the YIG/Pt control sample. Our findings provide a new insight into the spin current generation in Py and open an avenue in YIG-based spintronic devices.

## Results

### Structural and magnetic properties characterization
We epitaxially deposit a 10-nm YIG film with perpendicular magnetic anisotropy (PMA-YIG) on the (111)-oriented neodymium gallium garnet $Nd_3Ga_5O_{12}$ (NGG) substrate by magnetron sputtering (see Methods). To diminish the pinning effect of the PMA-YIG magnetization from the substrate, 0.5-nm Pt is sputtered as the buffer layer. Figure 1c, d present the out-of-plane and in-plane hysteresis loops of NGG/Pt(0.5 nm)/YIG(10 nm) obtained with a vibrating sample magnetometer (VSM),

respectively. They show a square out-of-plane loop and a typical hard-axis in-plane loop, indicating a perpendicular magnetic anisotropy. Figure 1e presents the X-ray diffraction (XRD) curves of a bare NGG substrate and the NGG/Pt(0.5 nm)/YIG($t_{YIG}$) films with different YIG thicknesses. When $t_{YIG}$ < 90 nm, the measured (444) peak positions (marked by black arrows) of the films appear at a higher angle than the expected bulk YIG (444) peak position (marked by the dashed line), indicating a tensile strain which relaxes with increasing YIG film thickness. The hysteresis loops of 30, 60, and 90-nm YIG show in-plane magnetic anisotropy (see Supplementary Note 2), while only 10-nm YIG shows PMA. Similar to previous reports[45], we attribute the PMA of thin YIG films to the lattice-mismatch-induced tensile strain.

### Anomalous inverse spin Hall effect in Py
On top of the PMA-YIG(10 nm) film, we deposit an Ag(4 nm)/Py(6 nm) bilayer and pattern it into a 30-μm wide and 2-mm long strip by photolithography and $Ar^+$ milling. Ag is needed to reduce the magnetic coupling between PMA-YIG and Py, and ensure the perpendicular anisotropy of PMA-YIG (see Supplementary Note 3). Subsequently, 300-nm insulating $SiO_2$ film and a metallic thin-film heater, Ti(10 nm)/Cu(30 nm)/Ti(20 nm), are deposited (Fig. 2a and Supplementary Note 4). A vertical temperature gradient can be generated when an AC charge current flows in the heater, driving a pure spin current flow from YIG into the Py layer. Wherein, the spin-to-charge conversion occurs and can be detected as the second harmonic signal with a lock-in amplifier[46].

Figure 2b shows the measurement geometry with the magnetic field along the $y$-direction. In this geometry, both YIG and Py magnetization are along the $y$-direction when a magnetic field with sufficient strength is applied. The thermoelectric signal along the $x$-direction comes from the conventional ISHE as well as the anomalous Nernst effect (ANE) in Py. Figure 2c presents the measured thermoelectric voltage $U$ in PMA-YIG(10 nm)/Ag(4 nm)/Py(6 nm) when the magnetic field is swept along the $y$-direction. We find $U$ has two parts: one shows a sharp switch at a low field, and the other one has a gradual change with the magnetic field and saturates at ~150 mT. To unveil the origins of these two parts, we compare the thermal voltage loop with the in-

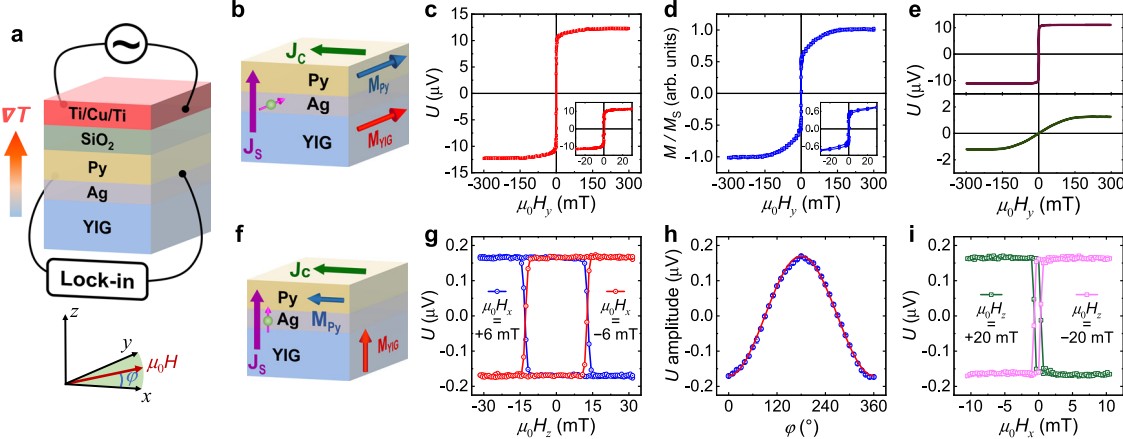

**Fig. 2 | Inverse spin Hall effect and anomalous inverse spin Hall effect measurements in PMA-YIG/Ag/Py trilayer. a** Schematic diagram of the thermally excited spin-to-charge conversion measurement. The AC current is applied in the Ti/Cu/Ti heater, and the thermoelectric signal in the Ag/Py bilayer is measured by a lock-in amplifier. **b** Sketch of the measurement geometry of ISHE and ANE. **c** The measured thermoelectric signal with the magnetic field swept along the $y$-direction. **d** In-plane magnetic hysteresis loop of PMA-YIG/Ag/Py with the magnetic field along the $y$-direction measured by VSM. The insets in **c**, **d** are the enlarged images at a small field range, respectively. **e** Decomposed two components in **c**. **f** Sketch of the measurement geometry of AISHE. **g** Measured thermoelectric signal with the magnetic field swept along the $z$-direction and a constant 6-mT magnetic field applied along the $+x$ (blue) and the $-x$ (red) direction. **h** Measured AISHE amplitude (calculated as the half of the difference between the voltages measured at the positive/negative maximum magnetic fields) as the function of $\varphi$ (the angle between the in-plane constant magnetic field and the strip as indicated in **a**). Symbols are the experimental data, and the line is the fitting with a $\cos\varphi$ function. **i** Measured thermoelectric signal with the magnetic field swept along the $x$-direction and a constant 20-mT magnetic field applied along the $+z$ (green) and the $-z$ (pink) directions, respectively.

plane magnetic hysteresis loop of the PMA-YIG/Ag/Py continuous film (Fig. 2d) and find they are very similar. For the in-plane PMA-YIG/Ag/Py loop, the magnetization switching at the low field corresponds to the Py layer, and the gradual change comes from the hard-axis loop of the PMA-YIG. Similar saturation fields in YIG/Ag/Py and bare YIG indicate small coupling between the two magnetic layers. Thus, we decompose the measured thermal signal in Fig. 2c as two components (Fig. 2e). The one with an easy-axis corresponds to the ANE of Py, and the other one with hard-axis characteristics represents the spin current from PMA-YIG induced spin-to-charge conversion in Py via the conventional ISHE. Here, the spin current $\mathbf{J_S}$ ($z$-direction), spin polarization direction $\hat{\sigma}$ ($y$-direction), and charge current $\mathbf{J_C}$ ($x$-direction) are orthogonal to each other. The difference between Fig. 2c, d is due to the different scaling factors for ANE and ISHE with the corresponding magnetization as be detailed in Supplementary Note 5.

To illustrate the collinear spin current generated spin-to-charge conversion, i.e., the AISHE in Py, we also perform similar measurements with the magnetic field swept along the $z$-direction (Fig. 2f). In this geometry, both spin current propagating direction and spin polarization align along the $z$-direction, forming a collinear spin current. The AISHE signal in Py can be detected when its magnetization is aligned along the $x$-direction. To do this, we apply a constant magnetic field $\mu_0 H_x = 6$ mT along the $+x$-direction to fix the Py magnetization while sweeping the magnetic field along the $z$-direction to flip the PMA-YIG magnetization. The thermoelectric voltage $U$ from AISHE in Py exhibits an easy-axis behavior, and changes polarity by reversing the PMA-YIG magnetization (blue loop in Fig. 2g). An opposite thermoelectric loop appears when the Py magnetization is aligned along the $-x$-direction (red loop in Fig. 2g). By rotating the in-plane constant magnetic field direction, we find the AISHE signal exhibits a $\cos\varphi$ dependence (Fig. 2h), where $\varphi$ is defined as the angle between the strip and the in-plane magnetic field. This is consistent with the predicted AISHE scenario[44], in which the spin-to-charge conversion signal is expected to be proportional to the component of Py magnetization along the $x$-direction. In addition, we also perform the thermoelectric measurement by fixing YIG magnetization along the $\pm z$-direction, while sweeping the magnetic field along the $x$-direction to flip the Py magnetization (Fig. 2i). In this geometry, the measured AISHE signal

coincides with the in-plane hysteresis loop of Py and changes sign when the magnetization of YIG switches. The measurement conditions of ISHE in Fig. 2c and AISHE in Fig. 2g–i are the same except for the different applied magnetic field directions. These results are fully consistent with the picture of the AISHE in a ferromagnetic metal[44] and in sharp contrast with the conversional ISHE where the spin-charge conversion is independent of the Py magnetization. Comparing these thermoelectric signals, we find the spin-to-charge conversion via the AISHE (Fig. 2g–i) is ~13% of that caused by the conventional ISHE (lower panel of Fig. 2e).

## Current-induced field-free switching of PMA-YIG

Above, we show that a perpendicularly flowing spin current with either $\hat{\sigma}_y$ or $\hat{\sigma}_z$ polarization can generate an in-plane charge current along the $x$-direction in Py, which originates from the ISHE and AISHE, respectively. Conversely, one can expect that an in-plane flowing charge current along the $x$-direction in Py with the magnetization along the $x$-direction can produce a spin current flowing along the $z$-direction with both $\hat{\sigma}_y$ and $\hat{\sigma}_z$ polarization via the SHE and ASHE, respectively. Thus, Py can be used as not only a conventional spin current source but also a collinear spin current source to realize the field-free switching of PMA materials. To demonstrate this, we etch the PMA-YIG/Ag/Py continuous film into a 100-μm long and 40-μm wide strip and detect its current and magnetic field induced perpendicular magnetization switching with polar magneto-optical Kerr effect (P-MOKE) (Fig. 3a), which is sensitive to the perpendicular magnetization. The magnetic field-dependent P-MOKE signal exhibits a square loop (Fig. 3b), which is very similar to the loop measured by VSM (Fig. 1c).

We next perform the current-induced switching of the PMA-YIG with a charge current and different external fields of $\mu_0 H_x$ along the $x$-direction. Figure 3c presents the P-MOKE signal with $\mu_0 H_x$ varying from $+10$ mT to $-10$ mT. The device exhibits deterministic anti-clockwise switching from $+10$ mT to $0$ mT. Remarkably, this also includes the case of zero magnetic field (the blue loop). Namely, a deterministic field-free switching is established. The field-free switching ratio is about 18%. The SOT-induced switching reverses its direction from anti-clockwise to clockwise when $\mu_0 H_x < -0.4$ mT. When $\mu_0 H_x$ increases from $-10$ mT to $+10$ mT, the loops are initially clockwise switching but

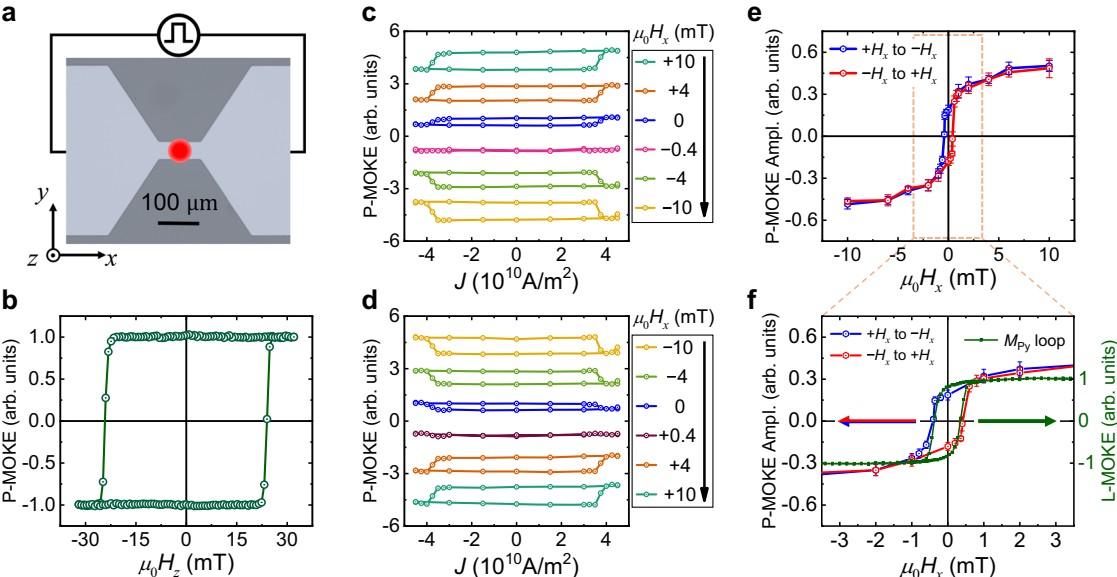

**Fig. 3 | SOT-switching of PMA-YIG/Ag/Py trilayer. a** Optical micrograph of the etched strip and the schematic electrical connection. The black region is the substrate and the gray marks the patterned trilayer. The red circle represents the laser spot used for the MOKE measurements. **b** Out-of-plane hysteresis loop of PMA-YIG/Ag/Py measured by P-MOKE. **c**, **d** are the SOT-switching of PMA-YIG/Ag/Py results with different $\mu_0 H_x$ varying from +10 mT to −10 mT and from −10 mT to +10 mT, respectively. **e** SOT-switching loop amplitude (Ampl.) of PMA-YIG/Ag/Py with different $\mu_0 H_x$. **f** Zoomed-in data of the orange block in Fig. 3e. The green line is the hysteresis loop of Py measured by L-MOKE when the DC charge current density is $3.25 \times 10^{10}\,\text{A/m}^2$. The detailed normalization process for P-MOKE is provided in Supplementary Note 7.

reverse to anti-clockwise switching when $\mu_0 H_x > 0.4$ mT (Fig. 3d). The field-free switching observed in PMA-YIG/Ag/Py is in sharp contrast to the current-induced magnetization switching in PMA-YIG/Pt bilayer, where no switching occurs at zero field and the switching polarity reverses when $\mu_0 H_x$ is reversed (see the next section). The $\mu_0 H_x$ dependent P-MOKE amplitude, which is defined as half of the difference between P-MOKE signals at positive and negative saturation current, exhibits a hysteresis-like behavior (Fig. 3e). Figure 3f presents the zoomed-in data of Fig. 3e at low fields and an in-plane hysteresis loop of Py measured by longitudinal magneto-optical Kerr effect (L-MOKE), under a DC current with $3.25 \times 10^{10}\,\text{A/m}^2$, a density around the switching threshold. The hysteresis-like feature of SOT-switching of PMA-YIG illustrates that the ASHE-induced $\hat{\sigma}_z$ polarization is proportional to the magnetization of Py along the x-direction. It shows a Py magnetization-controlled feature, and the field-free switching only occurs when Py has non-zero remnant magnetization along the strip. To further confirm this, we perform similar measurements after demagnetizing Py layer and do not find any field-free switching (see Supplementary Note 6). The finding of the correlation between the switching ratio with the remnant magnetization of Py suggests that it could be further improved through increasing the Py remanence.

**Spin-to-charge conversion and SOT-switching of PMA-YIG/Pt**
To further demonstrate the unique feature of the collinear spin current ($\hat{\sigma}_z$ polarization) in Py and its induced field-free switching of PMA-YIG, we compare it with the spin-to-charge conversion and SOT-switching measurements of a control sample: PMA-YIG/Pt. Figure 4a presents the typical ISHE signal in PMA-YIG(10 nm)/Pt(6 nm) bilayer. When the magnetic field is applied along the y-direction, the ISHE signal in the Pt layer depicts the y-component of YIG magnetization. Thus, it is similar to the in-plane hysteresis loop of PMA-YIG as presented in Fig. 1d. On the contrary, sweeping the magnetic field with the bias field of either $\mu_0 H_x = \pm 6$ mT (Fig. 4b) or $\mu_0 H_z = \pm 20$ mT (Fig. 4c) gives a negligible spin-to-charge contribution. These results are in sharp contrast to those observed in the PMA-YIG/Ag/Py trilayer (Fig. 2g, i), confirming

that only the y-polarized spin current can be converted into a charge current along the x-direction in Pt.

Figure 4d presents the out-of-plane hysteresis loop of PMA-YIG/Pt with P-MOKE, which is almost the same as that of PMA-YIG/Ag/Py system. Different from the deterministic switching at zero field in PMA-YIG/Ag/Py, we find that magnetization switching only occurs with the aid of an in-plane magnetic field for PMA-YIG/Pt (Fig. 4e). And the switching ratio increases with increasing magnetic field and saturates at ~6 mT (Fig. 4f). The sharp contrast between the SOT-induced switching of PMA-YIG/Pt and PMA-YIG/Ag/Py indicates the critical role of the collinear spin current with z-polarization in realizing the field-free perpendicular magnetization switching. Remarkably, the current density threshold of SOT-switching in PMA-YIG/Ag/Py is only about half of that in PMA-YIG/Pt, indicating a higher efficiency of SOT-switching by the combined effect of SHE and ASHE in Py, especially considering that the spin Hall angle of Py is lower than Pt. The finding of current threshold reduction with the adding of collinear spin current is in good agreement with previous theoretical prediction[47]. Therefore, we demonstrate field-free magnetization switching not only with a heavy-metal-free material, Permalloy (Py), but also with a higher efficiency as the current density threshold is only about half of that when Pt is used. The current threshold reduction cannot be explained by the field-free switching with stray field[17] or Néel orange-peel effect[40–43], indicating the importance of the collinear spin current in Py (see Supplementary Note 8).

## Discussion
In the conventional SHE, the relation of the spin current and charge current can be described by $\mathbf{J_S} = \frac{\hbar}{2e}\theta_{\text{SHE}}\mathbf{J_C} \times \hat{\boldsymbol{\sigma}}$, where $\theta_{\text{SHE}}$ represents the spin Hall angle. And the ISHE can be described by $\mathbf{J_C} = \frac{2e}{\hbar}\theta_{\text{SHE}}\mathbf{J_S} \times \hat{\boldsymbol{\sigma}}$. Due to the orthogonal relation between $\mathbf{J_C}$, $\mathbf{J_S}$ and $\hat{\boldsymbol{\sigma}}$, if a charge-to-spin conversion, $\mathbf{J_C} \to \mathbf{J_S}$ is via SHE, the inverse process would be $\mathbf{J_S} \to -\mathbf{J_C}$ with the same spin polarization. In the ASHE[44], a charge current in a ferromagnet can generate both orthogonal spin current and collinear spin current, depending on whether the applied charge current is orthogonal or parallel to the magnetization. And in the AISHE, a

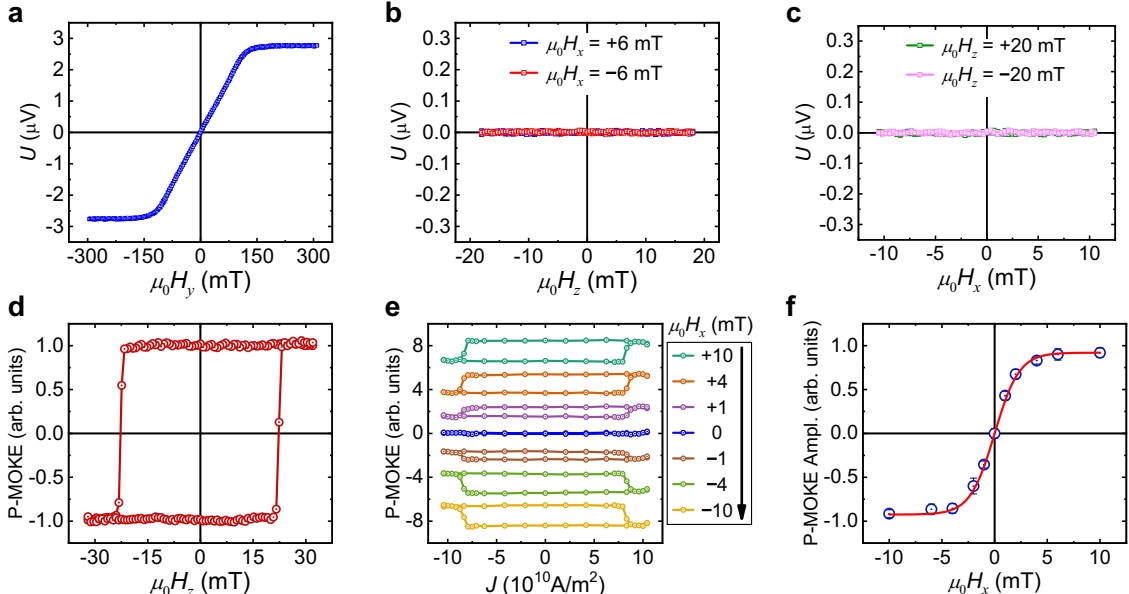

**Fig. 4 | Spin-to-charge conversion and SOT-switching measurements of PMA-YIG/Pt bilayer. a** Measured thermoelectric signal in Pt layer with the magnetic field swept along the $y$-direction. **b** Measured thermoelectric signal in Pt layer as a function of the magnetic field when it sweeps along the $z$-direction with a constant 6-mT magnetic field applied along the $+x$ (blue) and $-x$ (red) direction. **c** Measured thermoelectric signal in Pt layer as a function of the magnetic field when it sweeps along the $x$-direction with a constant 20-mT magnetic field applied along the $+z$ (green) and $-z$ (pink) direction. **d** Out-of-plane hysteresis loop of PMA-YIG/Pt measured by P-MOKE. **e** The current-driven SOT-switching of PMA-YIG/Pt at different $\mu_0 H_x$. **f** SOT-switching loop amplitude (Ampl.) of PMA-YIG/Pt as a function of $\mu_0 H_x$.

collinear spin current in a ferromagnet can generate charge current along the magnetization component and perpendicular to the spin current. Moreover, an orthogonal spin current in the ferromagnet can generate a charge current along the spin-current propagation (polarization) direction, whose magnitude is proportional to the magnetization component along the spin-current polarization (spin-current propagation)[44]. For the collinear spin current part of ASHE, the charge-to-spin conversion can be described by $\mathbf{J_S} = \frac{\hbar}{2e}\theta_{ASHE}\mathbf{J_C} \times (\hat{\boldsymbol{\sigma}} \times \hat{\mathbf{m}}_{Py})$, where $\theta_{ASHE}$ represents the anomalous spin Hall angle and $\hat{\mathbf{m}}_{Py}$ represents the direction of the Py magnetization. Its inverse process yields $\mathbf{J_C} = -\frac{2e}{\hbar}\theta_{ASHE}\mathbf{J_S} \times (\hat{\boldsymbol{\sigma}} \times \hat{\mathbf{m}}_{Py})$, where the negative sign is necessary to satisfy the Onsager relation[27]. Therefore, for a fixed spin polarization and Py magnetization, $\mathbf{J_S}$ and $\mathbf{J_C}$ are interlocked. Namely, if a charge-to-spin conversion $\mathbf{J_C} \to \mathbf{J_S}$ is via ASHE, the inverse process would be $\mathbf{J_S} \to \mathbf{J_C}$ via AISHE. Our results unambiguously confirm the distinct reciprocal relationships of charge-to-spin and spin-to-charge processes in the conventional SHE/ISHE and ASHE/AISHE. As the magnetization favors the up-state in both PMA-YIG/Ag/Py under the positive $\hat{\mathbf{m}}_{Py}$ (Fig. 3c) and PMA-YIG/Pt under the positive $\mu_0 H_x$ (Fig. 4e) when charge current flows along the $+x$-direction, the sign of spin-to-charge is negative for PMA-YIG/Ag/Py under positive $\hat{\mathbf{m}}_{Py}$ and positive $\mu_0 H_z$ (Fig. 2g) but positive for PMA-YIG/Pt under positive $\mu_0 H_y$ (Fig. 4a).

In summary, we find a ferromagnetic metal, Py, can efficiently convert a charge current to a collinear spin current and vice versa. We ascribe the spin-charge interconversion in Py to the recently proposed anomalous spin Hall effect and anomalous inverse spin Hall effect. With the unconventional spin current with the $z$-polarization in Py, we realized a current-induced field-free switching of perpendicularly magnetized insulator YIG. The magnetization switching of the PMA-YIG follows the hysteresis loop of Py along the $x$-direction. Remarkably, the field-free switching is achieved not only with a non-heavy metal, Py, as the spin current source but also with higher efficiency, since the current density threshold is only about half of that when using Pt. Although we demonstrate the field-free switching for PMA-YIG, the breaking of magnetic symmetry in a perpendicular direction with the collinear spin current is a general concept. We believe it should be applicable to other

perpendicular magnetic materials, including magnetic insulators and magnetic metals. Our findings provide a new insight into spin current generation in Py and open an avenue in YIG-based spintronic devices.

## Methods

### Sample preparation and device fabrication

On the NGG (111) substrate, a 0.5-nm Pt buffer layer and YIG was deposited by DC and RF magnetron sputtering, respectively. The sample was then annealed at 800 °C for 1 h in the atmosphere. Afterwards, Ag(4 nm)/Py(6 nm)/SiO₂(6 nm) or Pt(6 nm) were deposited on the PMA-YIG film by DC (Ag, Py, Pt) and RF (SiO₂) magnetron sputtering, respectively. Here, the SiO₂ layer was used to prevent the oxidation of Py. During the depositing of Ag/Py/SiO₂ layers, a 10-mT magnetic field was applied to increase the coercive field of Py. For the spin-to-charge conversion measurements, Ti(10 nm)/Cu(30 nm)/Ti(20 nm) multilayer was further deposited onto the Ag(4 nm)/Py(6 nm) and Pt (6 nm) as the heater, and 300-nm thick SiO₂ was used to isolate the heater and other metallic layers.

### Spin-to-charge conversion measurement

An AC current with an amplitude of 40 mA and a frequency of 187 Hz was applied in the Ti/Cu/Ti heater. The induced second harmonic thermoelectric signal was detected by a lock-in amplifier, SR860.

### Current-induced SOT-switching measurement

A DC current pulse of 10-ms duration was injected into the sample while the magnetization of PMA-YIG was monitored by a homemade MOKE system. The laser in our MOKE system is continuous-wave and the power is 17 mW. The center wavelength is 450 nm, and the spot diameter is around 60 µm. The measured MOKE signal is sensitive to out-of-plane/in-plane magnetization in polar/longitudinal configuration. The polar magneto-optical Kerr effect signal was obtained with laser incident normal to the surface. The longitudinal magneto-optical Kerr effect signal was obtained with an oblique incident laser. All the measurements were conducted at room temperature.

## Reporting summary

Further information on research design is available in the Nature Portfolio Reporting Summary linked to this article.

## Data availability

All data supporting the findings of this study are available within the main text and the Supplementary Information file. The data that support the findings of this study are available from the corresponding authors upon reasonable request. Source data are provided with this paper.

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

## Acknowledgements

We acknowledge the financial supports from the National Key R&D Program of China (Grants No. 2022YFA1403601 and No. 2021YFB3502400), the National Natural Science Foundation of China (Grants No. 12274203, No. 92165103, No. 12274204, No. 12241402, No. 51971110, No. 12374122, and No. 12374113), and the Hong Kong Research Grants Council (Grant No. 16300522).

## Author contributions

M.Y., L.S., B.M., X.W. and H.D. conceived the idea. M.Y., Y.Z., J.C., K.H. and L.Y. fabricated the samples and characterized the properties by VSM and XRD. M.Y. and Y.Z. performed the SSE measurements and SOT-switching measurements. M.Y., L.S., B.M., X.W. and H.D. wrote the draft. M.Y., L.S., Y.Z., J.C., K.H., X.Y., Z.W., L.Y., H.N., T.J., G.C., B.M., X.W. and H.D. performed the data analysis and discussed the results. All the authors revised the manuscript.

## Competing interests

The authors declare no competing interests.
