## [Peer Review File · Nature Communications]

Highly efficient field-free switching of perpendicular yttrium iron garnet with collinear spin currentREVIEWER COMMENTS

Reviewer #1 (Remarks to the Author):

The authors assert that efficient field-free switching of perpendicular yttrium iron garnet using a collinear spin current can be achieved through the anomalous Spin Hall Effect (ASHE). Their contributions can be summarized as follows:

1. The authors successfully detected the ASHE signal in the PMA-YIG/Ag/Py system using the electrical measurement technique known as Inverse SHE.
2. After confirming the presence of ASHE, the authors employed MOKE measurements to demonstrate the feasibility of field-free switching of perpendicular yttrium iron garnet with a collinear spin current in the PMA-YIG/Ag/Py system. Notably, they found that the required current density for this switching process is lower compared to the YIG/Pt system.

This research not only advances our understanding of ASHE in fundamental science but also holds significant implications for practical applications, particularly in achieving highly efficient field-free magnetization switching. It is anticipated that this work will have a noteworthy and broad impact across multiple scientific disciplines.

However, to ensure the manuscript's completeness and clarity, several aspects require further attention and clarification before publication:

1. **Material Selection:** The authors should elaborate on the rationale behind choosing NiFe (Py) as the upper ferromagnetic layer and Ag as the non-magnetic metal layer. It would be valuable to discuss how these material selections relate to factors such as energy band structure, spin properties, or spin diffusion length. Additionally, providing suggestions for alternative material choices would enhance the manuscript.
2. **Device Design:** Given that Ag is a conductive metal, the authors should provide an explanation of the sample design strategies employed to minimize any undesirable contributions from Ag during electrical measurements. Including microscopic images of the sample would enhance readers' understanding of the experimental setup and design.
3. **Comparison of Magnetization Behavior:** The authors should address the differences in magnetization behavior between the thermal voltage loop (Fig. 2c) and the in-plane magnetic hysteresis loop of the PMA-YIG/Ag/Py continuous film (Fig. 2d). An explanation for the origin of these differences should be provided. Furthermore, including enlarged images of Fig. 2c and Fig. 2d under low magnetic fields would facilitate a more detailed analysis.
4. **Clarification of the Mechanism:**

(1). **Field-Free Switching in the Remnant State of Py:** In Fig. 3c and Fig. 3d, the authors have depicted the SOT-switching process in the PMA-YIG/Ag/Py trilayer. Notably, field-free switching is achieved after the external magnetic field is applied and gradually removed. It is worth considering whether these results suggest that the field-free switching referred to in this article effectively occurs in the remnant state of Py. In other words, field-free switching might rely on Py being magnetized prior to operation. If this interpretation is accurate, the authors should explicitly clarify this aspect in the description of field-free switching, such as in the abstract or within the main body, to provide readers with a comprehensive

understanding of the operational mode.

(2). Contribution of Py Magnetization in SOT-Switching: The authors should further elucidate and discuss the potential contribution of the remnant magnetization or magnetization of Py as a small auxiliary magnetic field that assists in SOT-switching. This observation seems to explain the correspondence between the P-MOKE amplitude during SOT-switching and the L-MOKE (Py) behaviors, as shown in Fig. 3f. Addressing this point in more detail would significantly enhance the clarity of the underlying mechanism.

(3). Comparative Analysis with PMA-YIG/Pt: If feasible, the authors should consider conducting a comparative analysis of SOT-switching between PMA-YIG/Ag/Py without prior magnetization and PMA-YIG/Pt. Such a comparison could provide insights into the influence of SOT-switching on ASHE and reveal any efficiency differences in the switching process, contributing to a more comprehensive understanding of the observed phenomena.

Reviewer #2 (Remarks to the Author):

With the rising of magnonics, efficient control on magnetization of magnetic insulators, typically low damping YIG films, by electric methods is desired. In this case, the classic spin-transfer torque driven by spin-polarized current is failed; the spin-orbit torque technique becomes a promising alternative. However, the conventional SOT technique needs an additional field to assist switching PMA magnetization; a field-free SOT mechanism applicable to magnetic insulators is eagerly demonstrated. In this sense, the study is timely. The study also adopted the well-established longitudinal spin Seebeck effect to clearly characterize both ordinary inverse spin Hall effect and the anomalous inverse spin Hall effect. This measurement is smartly designed to obtain the relative magnitude of the AISHE angle to the conventional ISHE angle even if the concrete temperature gradient is unknown. Finally, the authors argue the PMA magnetization of YIG has been switched by the interplay of σ_z and σ_y without the need of additional external field. The measurement data seem solid and the manuscript has well been organized and written. I have the following comments and concerns.

1. My main concern is the uniqueness of the current interpretation. The LSSE has shown the presence of both AISHE and ISHE. The field-free switching data can also be possibly explained by the introduction of the AISHE. The classic picture of the Type-Z SOT mechanism, σ_y and a dipolar field colinear with MNiFe, is argued inactive because of the inappropriate switching direction. This argument is questionable because the dipolar field imposed by NiFe on YIG can be antiparallel to MNiFe as usual but also can be parallel to MNiFe owing to some other reasons such as the Néel orange-peel effect. The mechanisms of ASHE and the classic Type-Z scheme should be further identified by more evidences.
2. Are the measurement conditions of the ISHE and AISHE by LSSE the same? If so, readers can obtain directly relative ratio between the AISHE angle to that of ISHE.
3. When showing the relation among J_c , J_s and σ , they should be presented in the vector forms.
4. For the PMA-YIG/Ag/Py system, the Ag (4nm) layer is used to decouple direct exchange coupling between YIG and Py in the LSSE measurement. Nevertheless, the Ag layer is not indispensable for the SOT switching especially as the Ag layer is detrimental to shunter remarkable portion of current from NiFe. Will the switching current density be further lowered when the Ag layer is removed for the sake of even lower energy consumption?

5. The switching degree of the field-free switching should be mentioned for clarity.

Reviewer #3 (Remarks to the Author):

The authors report on field-free switching of perpendicular YIG by the collinear spin current. Because heavy-metal-free, they use the permalloy instead of traditional Pt metal to achieve the field-free magnetization switching, as mentioned in the manuscript. The authors could observe and ascribe the collinear spin current to the anomalous spin Hall effect in Py can be worth publishing as a paper. Anyhow, the manuscript contains enough information to satisfy the criteria of Nature Communication. Even so, I would like the authors to consider the following points before it can be accepted for publication.

- (1) In line 89, because XRD method and setup are not mentioned, I suppose the XRD measurements were performed under in-house X-ray machine. If so, the word "high resolution" is misleading.
- (2) In line 90, are all the YIG film (different thickness include 10, 30, 60 and 90nm YIG?) shown PMA? And, is the coercivity of YIG film as function as thickness increased (or decreased)?
- (3) In line 110, what is thickness of YIG layer are authors used?
- (4) In line 160, Fig 2, are the Pt 0.5 nm buffer layer on the substrate in PMA-YIG/Ag/Py trilayer system as same as mention on Fig 1c and 1d? If so, is any ISHE and AISHE effect from Pt buffer layer?
- (5) In line 166, Fig2, why is the figure different of the Fig 2c and 2d near the 0 field?
- (6) In line 219, Fig3b, what is value of coercivity? Is the hysteresis loop symmetry or asymmetry? Are some of the MOKE signals contributed from the Py layer?
- (7) how do the authors normalize all of the P-MOKE spectra and what is the rationale of the normalization? The meaning of the variation of current intensity changes with the normalization procedure.
- (8) It would be of interest to the readers if the authors described the spin diffusion length in their samples.

REVIEWER COMMENTS

Reviewer #1 (Remarks to the Author):

The authors assert that efficient field-free switching of perpendicular yttrium iron garnet using a collinear spin current can be achieved through the anomalous Spin Hall Effect (ASHE). Their contributions can be summarized as follows:

1. The authors successfully detected the ASHE signal in the PMA-YIG/Ag/Py system using the electrical measurement technique known as Inverse SHE.
2. After confirming the presence of ASHE, the authors employed MOKE measurements to demonstrate the feasibility of field-free switching of perpendicular yttrium iron garnet with a collinear spin current in the PMA-YIG/Ag/Py system. Notably, they found that the required current density for this switching process is lower compared to the YIG/Pt system.

This research not only advances our understanding of ASHE in fundamental science but also holds significant implications for practical applications, particularly in achieving highly efficient field-free magnetization switching. It is anticipated that this work will have a noteworthy and broad impact across multiple scientific disciplines.

However, to ensure the manuscript's completeness and clarity, several aspects require further attention and clarification before publication:

General response #1:

We sincerely thank Reviewer #1 for the time and efforts spent in evaluating our manuscript as well as the positive assessment of our work. We are delighted to learn that Reviewer #1 thinks our work “not only advances our understanding of ASHE in fundamental science but also holds significant implications for practical applications” and comments it “will have a noteworthy and broad impact across multiple scientific disciplines”. In the following, we provide our point-to-point responses to all the raised questions/comments.

1. Material Selection: The authors should elaborate on the rationale behind choosing NiFe (Py) as the upper ferromagnetic layer and Ag as the non-magnetic metal layer. It would be valuable to discuss how these material selections relate to factors such as energy band structure, spin properties, or spin diffusion length. Additionally, providing suggestions for alternative material choices would enhance the manuscript.

Response #1-1

We thank Reviewer #1 for this excellent suggestion. We choose Py as the spin-charge interconversion layer based on two factors. First, Py has been reported to have considerable spin-charge conversion efficiency in several previous works [Phys. Rev. Lett. 111, 066602 (2013), Appl. Phys. Lett. 104, 202405 (2014), J. Appl. Phys. 117, 172603 (2015), Phys. Rev. B 101, 064412 (2020)]. It can be anticipated that the field-assisted switching current threshold from the spin Hall effect itself would not be too high. Second, Py is a magnetic material which may contain the anomalous spin Hall effect. The aid of anomalous spin Hall effect may further reduce the switching current threshold to a value smaller than the typical heavy metal Pt, as reported in our manuscript. A nonmagnetic layer, Ag, is used to magnetically decouple the PMA-YIG and Py, and ensure a high ratio of pure spin current can be transferred between them. Meanwhile, it also has a low spin-charge conversion efficiency [J. Appl. Phys. 117, 172603 (2015)], thus not complicating the important role of anomalous spin Hall effect in Py on the highly efficient field-free switching. Utilizing the spin Seebeck measurement, we compared Cu and Ag, two typical materials with long spin diffusion length [Phys. Rev. B 72, 014461 (2005), Phys. Rev. Lett. 99, 196604 (2007)] and found that Ag is a better choice as the spin current can be transferred more efficiently (Fig. R1).

Fig. R1. The spin Seebeck effect signal measured in YIG/X(X = Ag, Cu)/Pt and YIG/Pt. The units of the numbers in parentheses are nm. The YIG films are deposited on thermally oxidized silicon substrates by RF magnetron sputtering and annealing at 800°C for 1 hour in atmosphere. Ag, Cu

and Pt are deposited by DC magnetron sputtering. The heater is made from Ti(10 nm)/Cu(30 nm)/Ti(20 nm) and is insulated with the sample by 200-nm SiO₂. An AC current with an amplitude of 100 mA and a frequency of 187 Hz is applied in the heater.

Since Ag is also a material with high conductivity, it may increase the switching current threshold due to the shunting effect. In the future, Ag could be replaced with a material with high resistivity and high spin current conductivity. In such case, the switching current density could be further lowered. NiO could be a good candidate if the coupling between NiO and the other two ferromagnetic materials are weak.

We added a short discussion about these in the updated Supplementary Note 1 and Note 3 sections.

2. Device Design: Given that Ag is a conductive metal, the authors should provide an explanation of the sample design strategies employed to minimize any undesirable contributions from Ag during electrical measurements. Including microscopic images of the sample would enhance readers' understanding of the experimental setup and design.

Response #1-2

We thank Reviewer #1 for raising this interesting question and the excellent suggestion. We agree with the reviewer that adding Ag may increase the switching current threshold due to the shunting effect. On the other hand, the non-magnetic metal, Ag is needed to reduce the magnetic coupling between PMA-YIG and Py. As shown in Fig. R2, when Py is directly deposited on YIG(10 nm), the strong magnetic coupling between them changes the magnetic behavior of YIG and it no longer has large remnant magnetization along the perpendicular direction. After inserting 4-nm thick Ag, YIG film shows a square loop almost identical with the one without capping, indicating negligibly small coupling between YIG and Py. We added a sentence “Ag is needed to reduce the magnetic coupling between PMA-YIG and Py, and ensure the perpendicular anisotropy of PMA-YIG (see Supplementary Note 3)” in revised manuscript at line 113~114. The detailed discussion is provided in the updated Supplementary Note 3 section.

Fig. R2. **a** and **b** are the hysteresis loops of PMA-YIG(10 nm)/Py(6 nm)/SiO₂(6 nm) and PMA-YIG(10 nm)/Ag(4 nm)Py(6 nm)/SiO₂(6 nm) in out-of-plane measured by polar MOKE.

As suggested, we also provided a microscopic image of thermoelectric measurement (Fig. R3) and the related information in the updated Supplementary Note 4 section.

Fig. R3. The microscopic image of thermoelectric measurement. The greenish blue, light blue and dark blue parts are the heater, sample and substrate, respectively.

3. Comparison of Magnetization Behavior: The authors should address the differences in magnetization behavior between the thermal voltage loop (Fig. 2c) and the in-plane magnetic hysteresis loop of the PMA-YIG/Ag/Py continuous film (Fig. 2d). An explanation for the origin of these differences should be provided. Furthermore, including enlarged images of Fig. 2c and Fig. 2d under low magnetic fields would facilitate a more detailed analysis.

Response #1-3

We thank Reviewer #1 for this excellent suggestion. The thermal voltage loop (Fig. 2c in main

text) has two origins. One is the anomalous Nernst effect (ANE) from Py, which scales with M_y of Py. The other one is the inverse spin Hall effect (ISHE) in Py, which scales with M_y of YIG. Generally, the scaling factors for ANE and ISHE with the magnetization are different. This results in a slight difference between the thermal voltage loop (Fig. 2c in main text) and the magnetization loop (Fig. 2d in main text). After the deconvolution of their individual contributions, one can find that they indeed have one-to-one correspondences (Fig. R4). We added a sentence “The difference between Figs. 2c&2d is due to the different scaling factors for ANE and ISHE with the corresponding magnetization as be detailed in Supplementary Note 5.” in the revised main text at line 140~142. The detailed discussion is provided in the updated Supplementary Note 5 section.

Fig. R4. The comparison of the deconvoluted ANE and ISHE contributions in the thermal voltage loop (a) and the magnetization contributions from Py and YIG in the magnetization loop (b), respectively. Apparently, they show close similarity, indicating the validity of the method for the deconvolution.

We also follow the suggestion and insert the enlarged figures in the field range of ± 30 mT as the insets in Fig. 2, which is also shown in Fig. R5. We can find the loops in the insets have similar coercivity, corresponding to the switching of in-plane Py magnetization. The slightly different slopes beyond the coercive fields are originated from the different scaling factors for ANE and ISHE with the magnetization.

Fig. R5. The updated Figs. 2c and 2d. We insert the enlarged images of Figs. 2c and 2d with the magnetic field between ± 30 mT.

4. Clarification of the Mechanism:

(1). Field-Free Switching in the Remnant State of Py: In Fig. 3c and Fig. 3d, the authors have depicted the SOT-switching process in the PMA-YIG/Ag/Py trilayer. Notably, field-free switching is achieved after the external magnetic field is applied and gradually removed. It is worth considering whether these results suggest that the field-free switching referred to in this article effectively occurs in the remnant state of Py. In other words, field-free switching might rely on Py being magnetized prior to operation. If this interpretation is accurate, the authors should explicitly clarify this aspect in the description of field-free switching, such as in the abstract or within the main body, to provide readers with a comprehensive understanding of the operational mode.

Response #1-4(1)

We thank Reviewer #1 for pointing out this to us. The similarity of the loop of polar MOKE amplitude during SOT-switching and the longitudinal MOKE amplitude loop of Py (Fig. 3f) indeed indicates the field-free switching only occurs when the remnant magnetization of Py along the strip is not zero, as pointed out by Reviewer #1. To further confirm this, we performed similar switching measurements after the demagnetization of Py layer and did not find any field-free switching [Please see **Response #1-4(3)**]. We added a sentence “It shows a Py magnetization-controlled feature and the field-free switching only occurs when Py has non-zero remnant magnetization along the strip. To further confirm this, we perform similar switching measurements after demagnetizing Py layer and do not find any field-free switching (see Supplementary Note 6).” in line 221~225 of the revised main text.

(2). Contribution of Py Magnetization in SOT-Switching: The authors should further elucidate and discuss the potential contribution of the remnant magnetization or magnetization of Py as a small auxiliary magnetic field that assists in SOT-switching. This observation seems to explain the correspondence between the P-MOKE amplitude during SOT-switching and the L-MOKE (Py) behaviors, as shown in Fig. 3f. Addressing this point in more detail would significantly enhance the clarity of the underlying mechanism.

Response #1-4(2)

We thank Reviewer #1 for this excellent suggestion. It was reported in nano-pillar system (dimension $\sim 300\text{ nm} \times 100\text{ nm}$) that the stray field of an in-plane magnetized CoFeB could induce the field-free switching of perpendicular magnetized CoFeB [Adv. Electron. Mater. 6, 1901368 (2020)]. However, our Py is patterned in a $100\text{-}\mu\text{m}$ long and $40\text{-}\mu\text{m}$ wide strip with two large pads at both ends (Fig. 3a in main text). In this situation, the generated stray field is expected to be very small when the magnetization is along the strip direction. In addition, the stray field is antiparallel with the magnetization direction of Py. The switching polarity from stray field is thus opposite to that from ASHE in Py observed in Figs. 3c&3d(in main text). Because the ASHE in Py is proportional to M_x of Py, we observed the correlation between the P-MOKE amplitude during SOT-switching and the L-MOKE (Py) behavior. We added this discussion in the updated Supplementary Note 8 section.

(3). Comparative Analysis with PMA-YIG/Pt: If feasible, the authors should consider conducting a comparative analysis of SOT-switching between PMA-YIG/Ag/Py without prior magnetization and PMA-YIG/Pt. Such a comparison could provide insights into the influence of SOT-switching on ASHE and reveal any efficiency differences in the switching process, contributing to a more comprehensive understanding of the observed phenomena.

Response #1-4(3)

We thank Reviewer #1 for giving this helpful suggestion. We measured the SOT-switching loops of PMA-YIG(10 nm)/Ag(4 nm)/Py(6 nm) both after demagnetizing Py, and with magnetic fields along the x -direction, shown in Fig. R6. We find the SOT-switching cannot occur when Py is demagnetized, due to the absence of collinear spin current. In this sense, SOT-switching is similar in PMA-YIG/Ag/Py (demagnetized) and PMA-YIG/Pt with only conventional y -direction polarized spin current. We added the discussion about this in the updated Supplementary Note 6 section.

Fig. R6. SOT-switching loops of PMA-YIG(10 nm)/Ag(4 nm)/Py(6 nm) both after demagnetizing the Py layer and with magnetic field along x -direction

Reviewer #2 (Remarks to the Author):

With the rising of magnonics, efficient control on magnetization of magnetic insulators, typically low damping YIG films, by electric methods is desired. In this case, the classic spin-transfer torque driven by spin-polarized current is failed; the spin-orbit torque technique becomes a promising alternative. However, the conventional SOT technique needs an additional field to assist switching PMA magnetization; a field-free SOT mechanism applicable to magnetic insulators is eagerly demonstrated. In this sense, the study is timely. The study also adopted the well-established longitudinal spin Seebeck effect to clearly characterize both ordinary inverse spin Hall effect and the anomalous inverse spin Hall effect. This measurement is smartly designed to obtain the relative magnitude of the AISHE angle to the conventional ISHE angle even if the concrete temperature gradient is unknown. Finally, the authors argue the PMA magnetization of YIG has been switched by the interplay of σ_z and σ_y without the need of additional external field. The measurement data seem solid and the manuscript has well been organized and written. I have the following comments and concerns.

General response #2:

We are very grateful to Reviewer #2 for the time and efforts spent in evaluating our manuscript as well as the positive assessment of our work. We are delighted to learn that Reviewer #2 thinks our “study is timely, measurements are smartly designed, data seem solid and the manuscript has been well organized and written”. In the following, we provide point-to-point responses to all the raised comments and concerns.

1. My main concern is the uniqueness of the current interpretation. The LSSE has shown the presence of both AISHE and ISHE. The field-free switching data can also be possibly explained by the introduction of the AISHE. The classic picture of the Type-Z SOT mechanism, σ_y and a dipolar field colinear with M_{NiFe} , is argued inactive because of the inappropriate switching direction. This argument is questionable because the dipolar field imposed by NiFe on YIG can be antiparallel to M_{NiFe} as usual but also can be parallel to M_{NiFe} owing to some other reasons such as the Néel orange-peel effect. The mechanisms of ASHE and the classic Type-Z scheme should be further identified by more evidences.

Response #2-1

We thank Reviewer #2 for raising this interesting question. The Néel orange-peel effect was used to explain the field-free switching in the in-plane CoFeB/W/PMA-CoFeB sandwich structures [Sci. Rep. 8, 8144 (2018), Phys. Rev. B 100, 104441 (2019), Appl. Phys. Lett. 120, 122402 (2022),

APL Mater. 11, 111104 (2023)]. In the SOT-switching measurement of PMA-YIG/Pt in Fig. 4e (in main text), we find the current density threshold almost does not change with the external magnetic field along x -direction within ± 10 mT. The field-free SOT-switching achieved by Néel orange-peel effect can be considered to be approximately equivalent to the conventional SOT-switching with the magnetic field applied along the charge current direction. As the spin Hall angle of Py is smaller than that of Pt, it is expected that in PMA-YIG/Ag/Py, the current density threshold for SOT-switching achieved by Néel orange-peel effect should be larger than that in PMA-YIG/Pt, which is opposite to what we observed. According to previous theoretical prediction [Sci. Rep. 10, 1772 (2020)], the z -polarized collinear spin current is critical to reduce the SOT-switching current density threshold. Therefore, though the Néel orange-peel mechanism may also have its contribution, the reduced switching current density threshold as compared with that in PMA-YIG/Pt indicates that the field-free SOT-switching is mainly from the collinear spin current in Py.

We briefly mentioned this in the main text in line 268~271, and added the detailed discussion about Néel orange-peel effect in the updated Supplementary Note 8 section.

2. Are the measurement conditions of the ISHE and AISHE by LSSE the same? If so, readers can obtain directly relative ratio between the AISHE angle to that of ISHE.

Response #2-2

We thank Reviewer #2 for this interesting question. The measurement conditions of ISHE and AISHE are the same except the applied magnetic field directions are different. The relative ratio between the AISHE and ISHE is estimated to be ~13%, as mentioned at line 168 in the updated manuscript.

We added a sentence “The measurement conditions of ISHE in Fig. 2c and AISHE in Figs. 2g~i are the same except for the different applied magnetic field directions.” at line 163~164 in the revised manuscript.

3. When showing the relation among J_c , J_s and σ , they should be presented in the vector forms.

Response #2-3

We thank Reviewer #2 for pointing out this. We corrected them and checked the manuscript

throughout.

4. For the PMA-YIG/Ag/Py system, the Ag (4nm) layer is used to decouple direct exchange coupling between YIG and Py in the LSSE measurement. Nevertheless, the Ag layer is not indispensable for the SOT switching especially as the Ag layer is detrimental to shunt remarkable portion of current from NiFe. Will the switching current density be further lowered when the Ag layer is removed for the sake of even lower energy consumption?

Response #2-4

We thank Reviewer #2 for this important suggestion. We agree with the reviewer that Ag is not indispensable for the SOT-switching. On the other hand, Ag is needed for removing the direct exchange coupling between Py and PMA-YIG, as also pointed out by the reviewer. As shown in Fig. R7, when Py is directly deposited on YIG(10 nm), the hysteresis loop along the perpendicular direction is no longer square and has a small remanence only. In the future, Ag could be replaced with a material with high resistivity and high spin current transparency. In such case, the switching current density could be further lowered. NiO could be a good candidate if the magnetic coupling between NiO and the other two ferromagnetic materials are weak. We added this discussion in the updated Supplementary Note 3 section.

Fig. R7. **a** and **b** are the hysteresis loops of PMA-YIG(10 nm)/Py(6 nm)/SiO₂(6 nm) and PMA-YIG(10 nm)/Ag(4 nm)/Py(6 nm)/SiO₂(6 nm) in out-of-plane measured by polar MOKE.

5. The switching degree of the field-free switching should be mentioned for clarity.

Response #2-5

We thank Reviewer #2 for this suggestion. The field-free switching degree at remanence state is 18%. Since it is correlated with the remnant magnetization, it could be further improved through increasing the Py remanence. We added this discussion in the revised manuscript at line 207 and line 225~227.

Reviewer #3 (Remarks to the Author):

The authors report on field-free switching of perpendicular YIG by the collinear spin current. Because heavy-metal-free, they use the permalloy instead of traditional Pt metal to achieve the field-free magnetization switching, as mentioned in the manuscript. The authors could observe and ascribe the collinear spin current to the anomalous spin Hall effect in Py can be worth publishing as a paper. Anyhow, the manuscript contains enough information to satisfy the criteria of Nature Communication. Even so, I would like the authors to consider the following points before it can be accepted for publication.

General response #3:

We sincerely appreciate the Reviewer #3 for the time and efforts spent in evaluating our manuscript as well as the positive assessment of our work. We are pleased to learn that the Reviewer #3 thinks our “manuscript contains enough information to satisfy the criteria of Nature Communication”. In the following, we provide our point-to-point responses to all the raised questions and concerns.

(1) In line 89, because XRD method and setup are not mentioned, I suppose the XRD measurements were performed under in-house X-ray machine. If so, the word “high resolution” is misleading.

Response #3-1

We thank Reviewer #3 for pointing out this to us. The model of X-ray diffractometer we used is Bruker D8 Advance. We deleted the “high resolution” at line 89 in the revised manuscript.

(2) In line 90, are all the YIG film (different thickness include 10, 30, 60 and 90nm YIG?) shown PMA? And, is the coercivity of YIG film as function as thickness increased (or decreased)?

Response #3-2

We thank Reviewer #3 for this interesting question. In this work, we find that only 10-nm YIG shows PMA, while 30, 60, 90-nm YIG show in-plane anisotropy (Fig. R8). The in-plane coercive fields of 30, 60, 90-nm PMA-YIG are 3.8, 2.5, 2.3 mT, respectively. The thickness dependent spin reorientation transition is originated from the competition of the dipole interaction and the interfacial PMA caused by the out-of-plane compressive strain. We added these information in updated

Supplementary Note 2 section.

Fig. R8. Hysteresis loops of 30, 60, 90-nm YIG for **a**, **b** and **c**, respectively. Purple loops are the out-of-plane loops and blue loops are in-plane loops.

(3) In line 110, what is thickness of YIG layer are authors used?

Response #3-3

We thank Reviewer #3 for pointing out this. The thickness of YIG in the thermoelectric and SOT-switching measurements is 10 nm. We revised this sentence to “On top of the PMA-YIG(10 nm) film , we deposit a Ag(4 nm)/Py(6 nm) bilayer and pattern it into a 30- μm wide and 2-mm long strip by photolithography and Ar^+ milling” at line 111 in the revised manuscript.

(4) In line 160, Fig 2, are the Pt 0.5 nm buffer layer on the substrate in PMA-YIG/Ag/Py trilayer system as same as mention on Fig 1c and 1d? If so, is any ISHE and AISHE effect from Pt buffer layer?

Response #3-4

We thank Reviewer #3 for raising these interesting questions. Yes, the Pt 0.5-nm buffer layer on the substrate in PMA-YIG/Ag/Py trilayer system is the same as that mentioned on Figs. 1c and 1d. But this is no ISHE and AISHE effect from Pt buffer layer, because Pt(0.5 nm)/YIG(10 nm) bilayer is insulating (measured with two contacts placed at the two diagonal ends of a 5 mm \times 5 mm sample). Since the buffer layer is isolated from the top metallic layer, Ag/Py, there is no electric current flowing in it. Thus, its contribution to ISHE and AISHE can be excluded.

(5) In line 166, Fig2, why is the figure different of the Fig 2c and 2d near the 0 field?

Response #3-5

We thank Reviewer #3 for raising this. The thermal voltage loop (Fig. 2c in main text) has two origins. One is the anomalous Nernst effect (ANE) from Py, which scales with M_y of Py. The other is the inverse spin Hall effect (ISHE) in Py, which scales with the M_y of YIG. The scaling factors for ANE and ISHE are different. This results in a slight difference between the thermal voltage loop (Fig. 2c in main text) and the magnetization loop (Fig. 2d in main text). After the deconvolution of their individual contributions, one can find that they indeed have one-to-one correspondences (Fig. R9). We added a sentence “The difference between Figs. 2c&2d is due to the different scaling factors for ANE and ISHE with the corresponding magnetization as be detailed in Supplementary Note 5.” in the revised main text at line 140~142. The detailed discussion is provided in the updated Supplementary Note 5 section.

Fig. R9. The comparison of the deconvoluted ANE and ISHE contributions in the thermal voltage loop (a) and the magnetization contributions from Py and YIG in the magnetization loop (b), respectively. Apparently, they show close similarity, indicating the validity of the method for the deconvolution.

(6) In line 219, Fig3b, what is value of coercivity? Is the hysteresis loop symmetry or asymmetry? Are some of the MOKE signals contributed from the Py layer?

Response #3-6

We thank Reviewer #3 for raising these interesting questions. The coercivity of Fig. 3b is ~ 24

mT. The coercivity fields at positive and negative fields are the same. The P-MOKE measured hysteresis loop depicts M_z of YIG, thus is anti-symmetry with magnetic field. In the P-MOKE geometry, the laser is normal to the sample surface. Thus, only the out-of-plane magnetization contributes to the signal. As the applied magnetic field is negligibly small compared to the demagnetization field of Py, the M_z of Py is also expected to be very small. In addition, P-MOKE signal from potential M_z of Py only serves as a linear background since the z -direction is the hard axis for Py, thus not affecting key information such as coercivity and saturation magnetic field.

(7) How do the authors normalize all of the P-MOKE spectra and what is the rationale of the normalization? The meaning of the variation of current intensity changes with the normalization procedure.

Response #3-7

We thank Reviewer #3 for raising this interesting question. In the MOKE, a linearly polarized laser strikes on the sample, then reflects off the sample through a polarizer and into a photodetector. When the sample is magnetized, the polarization direction of the light rotates, then the voltage measured by the photodetector (proportional to the light intensity) changes. In the hysteresis loop measured by P-MOKE (scanning H_z), we first subtract the linear background caused by the Faraday effect in the object lens. Then the remaining voltage difference for positively and negatively saturated states comes from the PMA-YIG only. The normalization actually involves two process. We first subtract the Kerr signal with the mean value of the measured loop. Then further normalize the Kerr signal to the saturation value at the positive field. In this way, we obtain the out-of-plane hysteresis loop of PMA-YIG measured by P-MOKE (such as Fig. 3b). For the SOT-switching loop measurements by P-MOKE (such as Figs. 3c and 3d), the magnetic field is constant thus Faraday effect-induced signal (from the lenses) does not change. We thus can directly normalize the saturated voltage difference for positive and negative currents in SOT-switching loop to the saturated voltage difference for positive and negative fields in magnetic hysteresis loop. Because the current intensity does not involve in the normalization process, the meaning of the variation of current intensity will not change. We added this in the updated Supplementary Note 7 section.

(8) It would be of interest to the readers if the authors described the spin diffusion length in their

samples.

Response #3-8

It was reported that the spin diffusion length of Ag and Py are 700 nm and 2.5 nm, respectively [Phys. Rev. Lett. 99, 196604 (2007), Phys. Rev. Lett. 111, 066602 (2013)]. As we are focusing on the field-free switching and Ag will provide the shunting effect, we did not measure the spin diffusion length of Ag. Our SSE measurements show the inserting of 4-nm Ag almost does not influence the transport of pure spin current (Fig. R1), indicating a long diffusion length of Ag. Our Py thickness dependent measurements yield the diffusion length of Py to be 4.2 ± 1.8 nm, in good agreement with previous estimated value of 2.5 nm. We mentioned this in the updated Supplementary Note 1 section.

REVIEWERS' COMMENTS

Reviewer #1 (Remarks to the Author):

The authors have thoroughly addressed my comments, significantly improving the quality of the manuscript. Consequently, I recommend that this work be accepted for publication in Nature Communications.

Reviewer #2 (Remarks to the Author):

The authors have replied to my and other referees' questions and comments properly. I am willing to recommend the publication of this paper in its current form.

Reviewer #3 (Remarks to the Author):

The authors present findings on field-free switching of perpendicular YIG induced by collinear spin current. Employing permalloy instead of traditional Pt metal, they achieve field-free magnetization switching, as detailed in the manuscript. The authors attribute the observed collinear spin current to the anomalous spin Hall effect in Py.

In the revised version, the author responded to pertinent questions from the review completely and augmented the manuscript with substantial additional information and references, thereby, it is anticipated that this work will have a high impact on science and industry application. These results hold promise for publication as a paper.